# Inclusion Complex between Local Anesthetic/2-hydroxypropyl-β-cyclodextrin in Stealth Liposome

**DOI:** 10.3390/molecules27134170

**Published:** 2022-06-29

**Authors:** Gredson Keiff Souza, André Gallo, Luiza Hauser Novicki, Heitor Rodrigues Neto, Eneida de Paula, Anita Jocelyne Marsaioli, Luis Fernando Cabeça

**Affiliations:** 1Chemistry Institute, State University of Campinas, UNICAMP, Rua Josué de Castro Cidade Universitária, Campinas CEP 13083-970, Brazil; gredsonkeiff@hotmail.com (G.K.S.); anita@iqm.unicamp.br (A.J.M.); 2Chemistry Department, Technological Federal University of Parana, UTFPR, Avenida dos Pioneiros, Londrina CEP 86036-370, Brazil; andre28gallo@gmail.com (A.G.); luizanovicki@alunos.utfpr.edu.br (L.H.N.); heitor_rneto@hotmail.com (H.R.N.); 3Biology Institute, State University of Campinas, UNICAMP, Rua Josué de Castro Cidade Universitária, Campinas CEP 13083-970, Brazil; depaula@unicamp.br

**Keywords:** butamben, ropivacaine, stealth liposomes, ^1^H-NMR, NMR-STD

## Abstract

The drugs delivery system in the treatment of diseases has advantages such as reduced toxicity, increased availability of the drug, etc. Therefore, studies of the supramolecular interactions between local anesthetics (LAs) butamben (BTB) or ropivacaine (RVC) complexed with 2-hydroxypropyl-β-cyclodextrin (HP-βCD) and carried in Stealth liposomal (SL) are performed. ^1^H-NMR nuclear magnetic resonance (DOSY and STD) were used as the main tools. The displacements observed in the ^1^H-NMR presented the complexion between LAs and HP-βCD. The diffusion coefficients of free BTB and RVC were 7.70 × 10^−10^ m^2^ s^−1^ and 4.07 × 10^−10^ m^2^ s^−1^, and in the complex with HP-βCD were 1.90 × 10^−10^ m^2^ s^−1^ and 3.64 × 10^−10^ m^2^ s^−1^, respectively, which indicate a strong interaction between the BTB molecule and HP-βCD (98.3% molar fraction and Ka = 72.279 L/mol). With STD-NMR, the encapsulation of the BTB/HP-βCD and RVC/HP-βCD in SL vesicles was proven. Beyond the saturation transfer to the LAs, there is the magnetization transfer to the hydrogens of HP-βCD. BTB and RVC have already been studied in normal liposome systems; however, little is known of their behavior in SL.

## 1. Introduction

Nuclear magnetic resonance (NMR) is one of the most important spectroscopic techniques for the investigation of supramolecular systems with interdisciplinary applications. NMR may promote evidence on the formation and topology of inclusion complexes formed by multiple components resulting from non-covalent hydrophobic interactions. The study of the different techniques based on ^1^H-NMR can promote evidence of a molecular self-assembly process, as well as the formation of a host–guest complex [1,2,3]. Such evidence is extracted from changes in chemical displacement, overhauser nuclear effect, relaxation, and diffusion coefficient measurements. In this context, ^1^H-NMR methods were developed and applied in the screening and characterization of supramolecular complexes such as: rotational nuclear frame Overhauser effect spectroscopy (ROESY 1D) [4] difusion-ordered spectroscopy (DOSY) [5] and saturation transfer difference (STD) [6].

Host molecules such as cyclodextrins (CDs) present in their structure a ring shape with a hydrophilic outer part, which ensures excellent aqueous solubility, and an internal apolar cavity, which can accommodate guest molecules. CDs allow the formation of stable inclusion complexes with a great diversity of organic substances [7,8]. CDs are composed of a group of pharmaceutical excipients composed of units of D-glucopyranose (cyclic sugars), of six, seven, and eight, called α-cyclodextrin (αCD), β-cyclodextrin (βCD) and γ-cyclodextrin (γCD). β-cyclodextrin is the most used, due to its favorable cavity size for medications [8,9]. It is noteworthy that cyclodextrins are not considered controlled drug release systems because they release the assets quickly as soon as they enter the bloodstream [9]. However, they are widely used to improve the solubility of insoluble compounds [7].

The carriers with excellent characteristics for controlled release are the liposome vesicles and they work as carriers (transporters) of drugs and biomolecules, regardless of their load or molar mass [9,10]. Usually, the liposomes are composed of phospholipids, that form stable concentric bilayers in an aqueous solution. Phosphatidylcholines present great stability in the face of pH or salt concentration variations in the medium [10,11].

The liposomes may also present modified membranes with polyethylene glycol originating long-lasting liposomes or stealth liposomes (SL). SL has a longer half-life time since it avoids the recognition and capture of these by the mononuclear phagocytic system due to the increase in vesicles solvatation [12]. The use of SL, in addition to increasing circulation time, also promotes an improvement in the stability of the complex and of the drug (by inhibiting degradation by enzymes and by reducing renal clearance of small molecules) [12] and decreased liposomal vesicle disintegration [13]. This is mainly because Poly(ethylene glycol) (PEG) is non-ionic, has low fouling, and has a high solubility in aqueous and organic media, which allows for the synthesis of PEGylated lipids, as well as facilitating the formulation of stealthy liposomes, which requires the solubilization of the PEG-lipopolymer [14].

Studies in the literature show that double drug encapsulation in cyclodextrins in liposomes can bring advantages such as better solubility of hydrophobic actives, inclusion complexes with controlled release properties, stability, and improvement to the integrity of its structure, reduction in toxicity and prolongation of drug time after being administered orally or parenterally [11,15,16].

An example of drugs encapsulated in liposomes that have been widely studied in the literature is local anesthetics (LA). Complexes formed with LA/liposomes show the advantages of the slow release of the drug, which prolongs the duration of anesthesia and reduction in toxicity to the cardiovascular and central nervous systems [11,17]. Great advances have been made in research involving liposomes and local anesthetics, which are promising in relation to the use of the free drug [18,19,20].

Ternary systems involving the anesthetic ropivacaine (RVC), HP-βCD, and liposomes were studied by Vieira et al. [11]. They observed that the anesthetic effect in the ternary system was able to prolong the RVC effect and the release of kinetics was also observed in cytotoxicity tests, thus reflecting a stronger interaction of RVC and HP-βCD. Moreover, local anesthetics such as ropivacaine and butamben (RVC, BTB) encapsulated in ternary systems open the prospect of promoting the potential long-term anesthesia with reduced cytotoxicity. However, it is of fundamental importance to have precise knowledge of the interaction between the compounds involved in the system, as well as the topology of the formed complex. Knowledge of this type of interaction can provide support for the development of more effective and potent anesthetics.

Hence, the present work focuses on developing a formulation for local anesthetic butamben (BTB) and ropivacaine (RVC) in a binary system, through its complexation in 2-hydroxypropyl-β-cyclodextrin (HP-βCD), followed by encapsulation in stealth liposomes. The study of the supramolecular interaction of different complexes was performed by nuclear magnetic resonance through spectroscopic analysis of ^1^H-NMR, DOSY and ^1^H-STD.

## 2. Results and Discussion

### 2.1. Subsection Complexing Efficiency (EE) and Constant Association (Ka) for BTB and RVC in HP-*β*CD

One of the objectives was to determine the encapsulation efficiency (EE) and affinity constant (Ka) of the complexes. UV-vis spectrophotometry was used for the anesthetic BTB and RVC.

For the EE and Ka calculations of local anesthetics RVC and BTB in HP-βCD, an anesthetic calibration curve was first performed. For the LA BTB, we presented the equation of equal line (Equation (1)) [7] and standard solubility data of BTB (S_0_) in buffer solution pH 7.4, S_0_ = 0.66 × 10^−3^ M for free BTB. Mura et al. [21] found the solubility of 0.86 mM.
Y(abs) = −0.0039 + 24.907 × [BTB] (R^2^ = 0.99884)(1)

As absorbance is an inherent property of each substance, in which different materials can absorb radiation at different wavelengths, absorption spectra were obtained in the UV region for both the determination of S_0_ and for the analytical curve of BTB (282 nm).

The absorbance values obtained from BTB at different concentrations of HP-βCD were used to find the value of the encapsulated BTB concentration, with the aid of the equation of the line of the calibration curve (Equation (1)) [22], as shown in Figure 1A.

The soluble phase of BTB in HP-βCD solution at pH 7.4 provided the equation of the line (Equation (2)). As shown in Figure 1A, it is noted that there is a formation of the binary complex of BTB and cyclodextrin, since the solubility diagram demonstrates that the drug has become more soluble as the concentration of HP-βCD increases [23]. The encapsulation efficiency (EE) of BTB in HP-βCD was 0.534, and it was calculated using the slope/1-slope equation (Equation (5)) [24,25]. According to Barbosa et al. [26] the higher the encapsulation efficiency, the lower the amount of cyclodextrin required for the solubilization of the drug, therefore, the closer it is to 1, and better the EE. Mura et al. [21] observed that the solubility of BTB increased linearly with the increase in the concentration of CDs, with the formation of highly soluble complexes in stoichiometry (1:1).
[BTB] = 0.34797 × [HP-βCD] + 1.19876 (R^2^ = 0.9997).(2)

The increase in the aqueous solubility of BTB in HP-βCD was used as a measure of complex formation. The value of the determined association constant (Ka) was Ka = 808 M^−1^ at pH 7.4 (where the S_o_ used was 0.66 × 10^−3^ M) Figure 1a. The value of the association constant (Ka) is used to compare the affinity of the drug with HP-βCD and also to classify the extent of chemical-physical changes that occurred after complexation. These values can range between 10 M^−1^ and 1000 M^−1^, and this unit is valid only for complexes with 1:1 stoichiometry [27]. In addition, Moraes et al. [28] point out that local amino-ester anesthetics present higher formation constant values, such as Ka = 549 M^−1^ for benzocaine and Ka = 351 M^−1^ for tetracaine. Thus, these results are in agreement with the literature. Maestrelli et al. [28] reported constant values of solubility of the order of 273 M^−1^ at pH 5. Moreover, this value depends both on the drug used and the medium in which it forms the complex with the carrier molecule. The value of the binding constant (Ka) is used to compare the affinity of the drug with HP-βCD.

The increase in total solubility of BTB in HP-βCD was 0.80 × 10^−3^ M, (about 21%); therefore, the solubility of the drug has a small increase in relation to the free BTB in an aqueous solution with pH 7.4. This result can be explained taking into account the use of low HP-βCD. High HP-βCD concentration values exhibit high values of encapsulated BTB absorption and eventually exceed detection limits [25]. In addition, the drug BTB already presents a good solubility even in the absence of the carrier molecule. Maestrelli et al. [28] reported BTB doubly loaded as a complex in liposome and hydroxypropyl-cyclodextrin; consequently, the complex showed good solubility and dissolution properties. The ratio BTB:HP-βCD was 1:4, which demonstrates that to form a 1:1 complex (HP-βCD:BTB) it is necessary four molecules of HP-βCD and a molecule of BTB in the solution [22,28].

For the EE and Ka calculations of the HP-βCD:RVC complex, a calibration curve of the RVC drugs (λ _=_ 263 nm) was first performed. For the RVC, we had the equation of the same line (Equation (3)) [7], where Y is the concentration of RVC and Abs absorption (R^2^ = 0.99945).
Y[_RVC]_ = −3.3586 × 10^−4^ + 0.0025 abs (3)

The absorbance values obtained from RVC at different concentrations of HP-βCD were used to find the value of the encapsulated RVC concentration, with the aid of the equation of the line of the calibration curve, as shown in Figure 1B.

The soluble phase of RVC in aqueous HP-βCD solution (Figure 1b) provided the equation of the line (Equation (4)) (R^2^ = 0.99942) [7]. The encapsulation efficiency (EE) of the RVC in HP-βCD was 0.210, and the value of the determined association constant (Ka) was 441 M^−1^ at pH 7.4 (Kd of 0.002 M), being S_0_ = 0.00048 M. The total solubility (St) of RVC in the presence of HP-βCD was 85 mM, 177 times higher than RVC in aqueous pH 7.4 solution. The RVC:HP-βCD ratio was 1:6. This result shows that in a 1:1 complex (RVC:HP-βCD) only one HP-βCD molecule in every six RVC molecules is forming the inclusion complex with RVC. However, for other local anesthetics, similarities in molar ratio were observed [11,23,29].
[RVC] = 0.1750 × [HP-βCD] + 0.00044 (4)

### 2.2. Analysis of LA/HP-*β*CD, LA/SL and LA/HP-*β*CD/SL Complexes Using Nuclear Magnetic Resonance (LA = BTB and RVC)

The characterization of the complexes was also performed by the spectroscopic technique of ^1^H-NMR. The first piece of evidence for the formation of a complex can be observed by varying the chemical shift in the ^1^H-NMR spectrum [1]. Figure 2 and Table 1 show the structures of the molecules of RVC, BTB, HP-βCD and the chemical shift of hydrogens, respectively.

For the RVC/SL complex, the highest values of shift variation were in (Δδ = 0.22) (H12; H11a); Δδ = 0.89 (H14). These values of chemical shift variation indicate that the cyclohexane ring of RVC is interacting with SL with greater intensity than the aromatic part of the molecule. The RVC/HP-βCD complex has little shift variation. For the RVC/HP-βCD/SL complex, there is also a low variation in the ^1^H RVC chemical shift, except for H12, H11a.

A small variation in the chemical shift of BTB in the inclusion complexes can be seen in Table 1. Variations of −0.12 for H3 and H5 in the BTB/HP-βCD and of −0.02 for H2 and H6 complex for BTB/HP-βCD/SL complex is also present. To prove the formation of the BTB/HP-βCD/SL complex encapsulation, efficiency tests of BTB were performed. EE values were found from 81.1% for the BTB/SL complex, and 84.2% for BTB/HP-βCD/SL; these values show that BTB is mostly encapsulated. According to the literature [29,30], the aromatic ring is expected in the cavity of the carrier, and interactions of the aromatic rings of BTB and RVC in the cavity of HP-βCD were observed.

Information regarding the variation in internal hydrogens of the HP-βCD cavity is important. This can happen due to the presence of molecules inside the cavity. An example is the ^1^H chemical shift variation in HC in the BTB/HP-βCD complex.

Another variation in HP-βCD hydrogens was in HA and HF. The hydrogens HA and HF (hydrogens near the lower cavity of HP-βCD) showed varying values in the chemical displacement of −0.03 and −0.06 ppm, respectively. This fact is due to the hydrogen bonds between the amino group or ester from BTB, with the HP-βCD hydroxide group. For the RVC/HP-βCD complex, the variation in the chemical shift of HP-βCD also was small. A possible topology for the inclusion of LA in HP-βCD was suggested in Figure 3.

For a better compression of binary and ternary inclusion complexes, ^1^H spectra of DOSY NMR were performed. From the diffusion coefficient data, the molar fraction (fx) and the association constant (Ka) of the complexes were calculated, as shown in Table 2.

For the RCV/SL complex, a molar fraction value of the 18% complex is found with its constant association 28 L mol^−1^, which proves the association of RVC and SL. The formation of inclusion complex in a solution containing a drug and cyclodextrin can be observed by reducing its diffusion coefficient (D). The greater the difference between D in solution with cyclodextrin compared to the D obtained from the solution without cyclodextrin, the greater the fraction of the drug inclusion complex [31].

The RVC/HP-βCD and RVC/HP-βCD/SL complexes indicated a complex fraction value of 22% and 24%, respectively; thus, approximately 25% of the RVC is complex. For the constant association, there is 37 and 44 L/mol of the RVC/HP-βCD and RVC/HP-βCD/SL complex, respectively. According to Araujo et al. [24] the typical chemical structure of LAs is characterized by a hydrophilic region (an amine group) and another hydrophobic region (usually an aromatic ring) separated by a polar group of the ester or amide type, which can generate differences in the proportion between the neutral and the charged form that is responsible for the speed of action. These characteristics present relatively low pKa values, between 7.6–8.9.

For the BTB/HP-βCD complex, the high molar fraction value of the obtained complex was 98.3%, jointly with its high constant association (72.279 × 10^3^ L mol^−1^), which proves the association of BTB and the HP-βCD molecule. The results corroborate the high values found for Ka for the Uv-vis experiment (Ka 808 M^−1^). Values of the BTB/SL and BTB/HP-βCD/SL complex were not reported because no BTB signals were not obtained in the complex spectrum. This is probably due to the fact that BTB is not in equilibrium between the aqueous phase and SL.

For better observation of the inclusion complexes between liposomes and the local anesthesia BTB or RVC, experiments of ^1^H NMR of saturation transfer differ (STD) were applied to study the local anesthetic-liposome interactions [32,33].

The STD experiment provides information on the hydrogens of the molecule BTB or RVC and HP-βCD that are encapsulated in the SL vesicle. Figure 4 indicates the STD spectrum for the BTB/HP-βCD/SL complex. Figure 4A refers to the ^1^H spectrum of the ternary complex. Figure 4B expresses the control spectrum (out of resonance) and Figure 4C the STD spectrum. In the region of 3 to 4 ppm, very intense signs were observed. They refer to the hydrogens of polyethylene glycol (PEG) and HP-βCD, which are overlapping; therefore, this region was discarded for STD analysis.

The STD spectrum (irradiated at −0.5 ppm) indicated that the liposome saturation transfer to BTB and HP-βCD hydrogens occurred; this indicates that both molecules were encapsulated in SL [32,33]. For the hydrogens signals of HP-βCD, it was not possible to calculate the transference of the magnetization percentage due to the overlap of the signals. For the BTB molecule, the aromatic hydrogens H2, H3, H5, and H6 presented the highest magnetization transfer rates (97%), followed by H8 (87.5%) as shown in Table 3.

The STD experiment also provided an analysis of the ternary complex RVC/HP-βCD/SL. This analysis provides information about the hydrogens of the RVC molecule and the HP-βCD that are encapsulated in the SL vesicle (Figure 5). Figure 5C refers to the ^1^H NMR spectrum of the ternary complex. It can be observed that the signs of RVC are enlarged due to the degree of interaction with the liposomal vesicle. Figure 5B shows the control spectrum (out of resonance) and Figure 5A the STD spectrum. In the region of 3 to 4 ppm, the signals are more intense; they refer to the HP-βCD hydrogens and PEG polymer (which coats the surface of the SL). Due to the overlap of signals, this region was discarded for STD analysis.

The RVC hydrogens with higher saturation transfer intensity were those of the aromatic ring 100% (H1, H2, H3), followed by H15–H16 and H14 34% and 35%, respectively. In the STD spectrum, H12 and H11a signals were also observed, but it was not possible to calculate the std transfer value because its intensity was not significant in the control spectrum. Thus, other signals show a greater intensity and overlap their maximum intensity. However, it can be considered that it has been complexed in the liposomal vesicle.

HP-βCD hydrogens also showed saturation transfer. Hydrogens in the region between 3–4 ppm were not considered due to the overlap of signals. HI presented 0.4% in saturation transfer, indicating the lowest transfer value, as shown in Table 3.

The SL saturation transfer values for RVC and HP-βCD hydrogens confirm the encapsulation of the RVC/HP-βCD inclusion complex in SL vesicles. The STD spectrum shows that the aromatic part presents greater interaction with the liposomal vesicle since the Hs of the N-propyl amine and piperidine ring are also interacting with the SL, although with lower intensity.

For a better understanding of the complex interactions involving local anesthetics, the spectroscopic technique of ^1^H STD and DOSY has been applied, which shows suggestions of topologies and possible interactions among the molecules involved in the complex. Cabeça et al. [33] studied the mixture of prilocaine/βCD/EPC at different pHs and reported that prilocaine molecules were incorporated into liposomes. With the aid of the ^1^H STD technique, it was observed that prilocaine was released from βCD to liposome vesicles, and the formation of the PLC-Liposomes complex was higher at pH 10. Martins et al. [34] observed a strong interaction between the antibiotic dapsone and egg phosphatidylcholine liposome (EPCL), with the complete insertion of dapsone in the lipid bilayer of the liposome. The authors reported that this type of system may be an alternative application for a low-soluble substance. In addition, embroidery may be useful for investigating other binary/ternary mixtures in solution.

### 2.3. Analysis of Particle Size and Zeta Potential

The analyses of particle size, polydispersity index (IP), and zeta potential (PZ) are shown in Table 4. Values of 200 ± 0.91 nm and 200 ± 0.88 nm for the binary complex of BTB/SL and RVC/SL and 229 ± 0.91 nm and 285 ± 0.91 nm for the ternary complex of BTB/HP-βCD/SL and RVC/HP-βCD/SL are indicated, respectively.

The results indicate that liposome vesicles were homogeneous even after the addition of BTB, RVC, and the HP-βCD:RVC complex with polydispersity index with lower values ≤ 0.4. This result points to a formulation close to homogeneity among SL, BTB, RVC, and HP-βCD.

The surface load of vesicles for both ternary and binary complexes was negative and close ranging from 22 to 25 (Table 4). According to Cavalcanti et al., [25] IP ranges from 0 for homogeneous to 1 for polydispersed samples, since it is a measure of the amplitude of the size distribution, that resulted from the cumulative analysis of dynamic light dispersion (DLS) data. An IP equal to 1 indicates large variations in particle size, and a value close to 0 indicates a population of monodispersed particles (Lacatusu et al., [35]).

## 3. Materials and Methods

### 3.1. Materials

1,2-dipalmitoyl-sn-glycero-3-phosphocholine (DPPC); 1,2-distearoil-sn-glycero-3-phosphoethanolamina N[methoxy(polyethylene glycol)]-2000) (DSPE-PEG-2000); cholesterol were acquired from Avanti^®^ polar lipids; Deuterated solventes; Ropivacaine (RVC) ((2S)-N-(2,6-Dimethylphenyl)-1-propyl-2-piperidinecarboxamide) and Butamben (BTB) Butyl 4-aminobenzoate, were acquired from Sigma Aldrich, São Paulo, SP, Brazil. 

### 3.2. Preparation of the Complex between Local Anesthetics RVC or BTB in HP-*β*CD

The inclusion complex BTB/HP-βCD or RVC/HP-βCD was prepared by mixing an equimolar amount of BTB and RVC in HP-βCD (1:1) in sodium phosphate pH buffer solution 7.4. The mixture was left on a stirrer table for 24 h and then filtered in polycarbonate membrane (millex filter, MilliporeSigma, Burlignton, MA, USA with 0.45 μm pore).

### 3.3. RVC or BTB Calibration Curve

For the preparation of the calibration curve, BTB samples were prepared in phosphate pH buffer solution 7.4, varying the concentration from 0.014 to 0.066 mM. In order to obtain the RVC curve form, samples of RVC were prepared in phosphate-buffered solution pH 7.4 with concentration ranging from 0.1 to 0.54 mM. After that, the samples were taken to UV-vis.

### 3.4. Preparation of BTB Solutions for the Determination of Maximum Solubility (S_0_) in Sodium Phosphate Buffer pH 7.4

For the determination of the maximum solubility of BTB in sodium phosphate buffer pH 7.4, the methodology available in the Brazilian Pharmacopoeia 5th Edition (ANVISA, 2010) was used. The preparation of the solutions consisted of the addition of excess drug (BTB) (5, 10, 20, 30, 40, 50 mM) to constant volumes of solvent (3 mL), aiming at obtaining a saturated solution. According to the procedure systematized by Pharmacopoeia, the total content of solute in UV-vis solutions was determined.

### 3.5. Complexing Efficiency and Phase Solubility of the BTB/HP-*β*CD and RVC/HP-*β*CD

The complex formed between BTB and HP-βCD was prepared in pH 7.4 buffer solution (sodium phosphate buffer 0.05 mM), varying the concentrations of HP-βCD 0.001 to 0.5 mM, with an excess BTB (150 mM). The solutions were left on an agitator table for 24 h. The samples were filtered in polycarbonate membrane (millex filter, Millipore, USA with 0.45 μm pore) and taken for UV-vis analysis. For the RVC and HP-βCD complex, the variation in HP-βCD concentration was 0.001 to 0.4 mM with excess RVC (70 mM). The mixture was stirred at room temperature for 24 h to achieve balance [25]. The sample was filtered in a membrane of 0.45 one (Millipore). The concentration of each LA for each sample was determined by UV spectrophotometry at 263 nm for RVC and 282 nm for BTB (Spectrometer Lambda 25). The association constant (Ka) was determined from the slope of the linear relationship between the molar concentrations of LA in soluble medium versus the molar concentration of HP-βCD according to Equation (5) [23]. The encapsulation efficiency (EC) of the LAs was determined from the data of the phase solubility curve and according to Equation (6) [22]. The drug/cyclodextrin ratio can be found in Equation (7) [7].
(5)$$K=\frac{slope}{S_{0}x\;\left(1-slope\right)}$$
(6)$$EE=\frac{slope}{1-slope}$$
(7)$$\mathrm{LA}:\mathrm{HP}\text{-}\beta \mathrm{CD}\;=1\;:\;1+\frac{1}{\begin{array}{c} EE \end{array}}$$
where *S*_0_ is the aqueous solubility of LA in the absence of HP-βCD.

### 3.6. Preparation and Encapsulation of LA in Long-Circulation Liposomes (RVC/SL and BTB/SL)

Liposomes were prepared using the lipid film method [36]. Stealth liposomes were formulated from egg phosphatidylcholine (EPC), cholesterol, and polyethylene glycol DSPE-PEG-2000 at a molar ratio of 54:41:5. Lipids were solubilized in chloroform containing 10 mM of RVC or BTB. The solvent was removed at room temperature to obtain a lipid film. Then, the lipids were hydrated with a pH 7.4 phosphate-buffered solution and stirred in the vortex. The complex was left in balance for 2 h. The formulation was then extruded 13 times in 400 nm polycarbonate membrane in an Avanti mini extruder.

### 3.7. Ternary Complex Preparation (RVC/HP-*β*CD/SL)-(BTB/HP-*β*CD/SL)

The stealth liposomes were prepared from the compounds of egg phosphatidylcholine (EPC), cholesterol (Sigma Aldrich, São Paulo, SP, Brazil), and DSPE-PEG-2000 (Avast lipids) at a molar ratio of 54:41:5. The components were solubilized in chloroform and left at room temperature to obtain a lipid film [37]. Then, the lipid film was hydrated with the binary complex of (RVC/HP-βCD) or (BTB/HP-βCD), taken to the vortex, and left in equilibrium for 2 h. The liposomal suspension formed multilamelares liposomes, which in turn were extruded into a 400 nm polycarbonate membrane in an Avanti mini extruder to obtain small unilamellares vesicles (SUV).

### 3.8. Determination of Encapsulation Efficiency of the LA/SL and LA/HP-*β*CD/SL Complex (SL = RVC or BTB)

The efficiency of LA encapsulated in liposomes was determined using the ultrafiltration/ultracentrifugation technique (Millipore, USA, ME Cut-off 10,000Da) [28]. A sample amount of 400 μL containing the LA/SL or LA/HP-βCD/SL complex was placed in a filter unit and subjected to ultracentrifugation (13,000 rpm for one hour). A quantity of 100 μL that passed through the filter (free LA) was diluted in 3 mL of phosphate buffer (pH 7.4) and quantified using UV spectrophotometry at 263 nm for RVC 282 nm for BTB. The efficiency calculation was performed using Equation (8) [28].
(8)$$EE\%=\frac{{\left[\mathrm{LA}\right]}_{total}-\left[\mathrm{LA}\right]{\;}_{diffused.AL}}{{\left[\mathrm{LA}\right]}_{total}}\times 100$$

### 3.9. Nuclear Magnetic Resonance

NMR experiments were carried out at the Laboratory of Nuclear Magnetic Resonance of the State University of Londrina—UEL. The equipment used was the Bruker 400 MHz for hydrogen frequency (Bruker Corporation, Billerica, MA, USA), carried out with “software” Bruker standards under typical conditions. The experiments were conducted at 25 °C, applying as reference the peak of the residual deuterium (4.70 ppm), which was used as field lock and adjusted the homogeneity of the magnetic field.

DOSY design: For all experiments, 16 different gradient amplitudes were used with diffusion time optimization of 0.06 s. The DOSY Toolbox [36] data processing program was employed. The coefficients calculated for each selected signal were listed together with the respective standard deviations. The value of the diffusion coefficient and standard deviation of each species involved in the analysis was given through the arithmetic mean of all coefficients of the same species. Coefficients with values different from those presented by the majority were discarded. The results of the DOSY analysis method are two-dimensional spectra with 1H NMR chemical displacements on one axis and the calculated diffusion coefficient (m^2^s^−1^ × 10^−10^) in another dimension. The mean time of acquisition of the experiment was 25 min. The complex diffusion coefficient is determined based on the exchange between a free and complex state of the ligand. The complexed molar fraction (*fx*) and association constant (*Ka*) can be calculated using Equations (9) and (10), respectively [23].
(9)$$fx=\frac{\left(D_{free}-D_{complex}\right)}{\left(D_{free}-D_{host}\right)}$$
(10)$$Ka=\frac{f_{x}}{\left(\left(1-f_{x}\right)\left(\left[host\right]-f_{x}\;\left[guest\right]\right)\right)}$$

STD experiment: The STD experiments were selectively saturated using Gaussian pulse trains at −0.5 ppm for resonance acquisition and 30 ppm for out-of-resonance acquisition. The experiments were processed using the TOP SPIN program. Saturation time was 2.55 s.

### 3.10. Determination of Size, Zeta Potential and SL Particle Polydispersivity Index

The average particle size, as well as zeta potential and liposome polydispersivity index were determined through dynamic light scattering experiments carried out at the Biology Institute of Unicamp. The determination was performed by an average of the data obtained in the experiments. The device used in the study was a Zeta Size particle analyzer from Malvern Instruments (Worcestershire, UK).

## 4. Conclusions

The molecular interactions of local anesthetics BTB and RVC with HP-βCD were identified by the DOSY technique and also by the ^1^H chemical shift, which the aromatic rings of the drugs interacted with the HP-βCD cavity. It was observed in the normalized ^1^H-STD that the BTB/HP-βCD complex showed variations in intensity of all ^1^H signals, with aromatics hydrogens showing the highest intensity values. HP-βCD ^1^H signals in the region between 3 and 4 ppm were also observed. This result can confirm the encapsulation of the binary complex in the liposomal vesicle. For the RVC anesthetic, lower values of encapsulation were observed both in the binary complex and in the liposomes (DOSY). The complex RVC/HP-βCD showed an alternative of topology whit the aromatic hydrogens and the piperidine ring in hydrophobic cavity of HP-βCD. For the ^1^H-STD of RVC/HP-βCD complex, the RVC hydrogens with higher saturation transfer intensity were those of the aromatic ring, and there are HP-βCD ^1^H signals in the region between 3 and 4 ppm too. In general, these results are promising, indicating that ^1^H NMR techniques for mapping the bonds intermolecular of complexes may be utilized. This opens the door to further studies, aiming at better systems for drug protection and release, since the field of study in supramolecular chemistry grows exponentially.

## Figures and Tables

**Figure 1 molecules-27-04170-f001:**
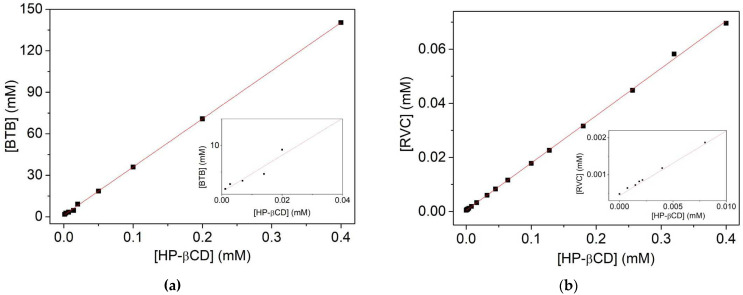
(**a**) BTB Solubility in HP-βCD (slope = 0.3479, R^2^ = 0.99977); (**b**) RVC Solubility Graphic In HP-βCD (slope = 0.175, R^2^ = 0.9994). Determined at room temperature (n *=* 3).

**Figure 2 molecules-27-04170-f002:**
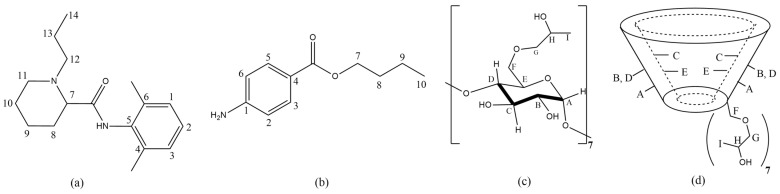
Structures of the molecules of (**a**) RVC; (**b**) BTB; (**c**,**d**) HP-βCD.

**Figure 3 molecules-27-04170-f003:**
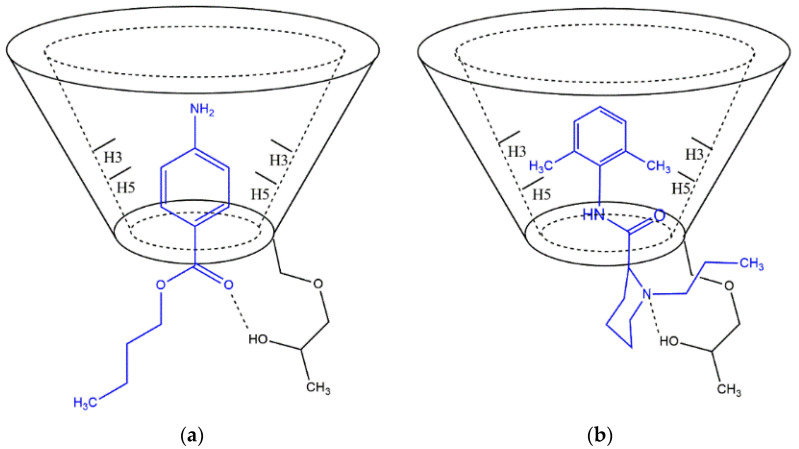
(**a**) Alternative topology for the BTB/HP-βCD and (**b**) RVC/HP-βCD complex.

**Figure 4 molecules-27-04170-f004:**
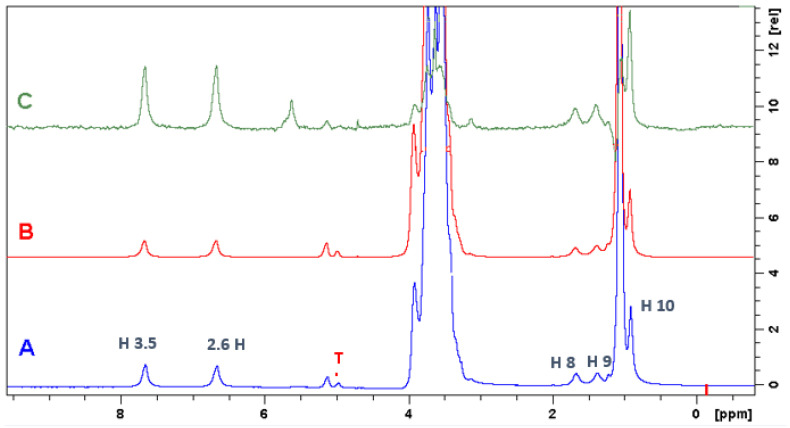
(**A**) ^1^H NMR spectrum (400 MHz, D_2_O/residual H_2_O reference at 4.7 ppm) of the BTB/HP-βCD/SL complex (50 mmol L^−1^); (**B**) control spectrum (radiating at 30 ppm); (**C**) STD spectrum (radiating at −0.5 ppm). Total saturation time 2.55 s.

**Figure 5 molecules-27-04170-f005:**
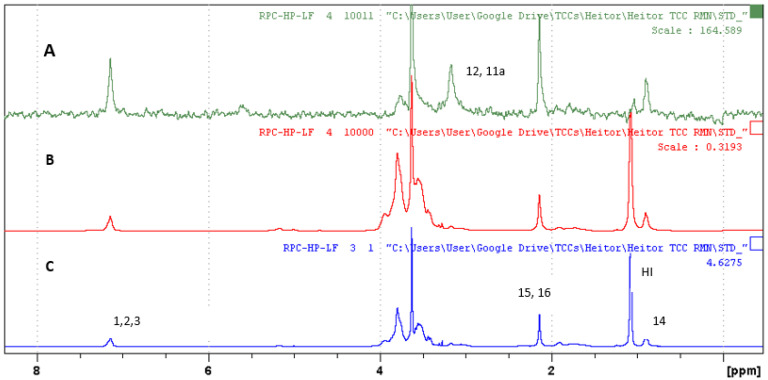
(**A**) STD spectrum (radiating at −0.5 ppm); (**B**) control spectrum (radiating at 30 ppm); (**C**) ^1^H NMR spectrum (400 MHz, D_2_O/residual H_2_O reference at 4.7 ppm) of RVC/HP-βCD/SL complex [50 mmol ^L−1^]. Total saturation time 2.55 s.

**Table 1 molecules-27-04170-t001:** Attribution and chemical shift of BTB ^1^H and RVC ^1^H in the binary and ternary complexes.

**H**	**BTB (ppm)**	**BTB/SL δ ^1^H (ppm)**	**Δδ (ppm)**	**BTB/HP-** **βCD (ppm)**	**Δδ (ppm)**	**BTB/HP-βCD/SL (ppm)**	**Δδ (ppm)**
2	6.76	-	-	6.69	−0.07	6.67	−0.02
3	7.78	-	-	7.66	−0.12	7.66	0.00
5	7.78	-	-	7.66	−0.12	7.66	0.00
6	6.76	-	-	6.69	−0.07	6.67	−0.02
7	4.23	-	-	4.26	0.03	-	-
8	1.66	-	-	1.68	0.02	1.68	0.00
9	1.36	-	-	1.39	0.03	1.38	0.01
10	0.87	-	-	0.91	0.04	0.92	0.01
**H**	**RVC * (ppm)**	**RVC/SL *** **δ ^1^H (ppm)**	**Δδ (ppm)**	**RVC/HP-βCD * (ppm)**	**Δδ (ppm)**	**RVC/HP-βCD/SL * (ppm)**	**Δδ (ppm)**
1,2,3 Aromatic	7.14	7.14	0.00	7.15	0.01	7.14	-
7	4.12	4.10	−0.02	4.10	−0.02	-	-
8 _e_	2.37	2.37	0.00	2.37	0	-	-
11 _e_	3.67	-	-	-	-	-	-
12, 11 _a_	3.08	3.30	0.22	3.06	−0.02	3.17	0.09
15, 16	2.12	2.12	0.00	2.14	0.02	2.14	0.02
14	0.89	0.89	0.89	0.9	−0.01	0.90	0.01

_a_ axial hydrogen; _e_ equatorial hydrogen; * concentration = 10 mM

**Table 2 molecules-27-04170-t002:** Diffusion coefficients (*D*) of LA (BTB, RVC), HP-βCD and LA/HP- βCD, LA/HP-βCD/SL constant association (*Ka*) and molar fraction (*fx*).

Complex	Compounds	*D* (10 × −10 m^2^ s^−1^)	Molar Fraction of *fx* % Complex	*Ka* L/mol
	RVC	4.07 ± 0.40		
	SL	0.32 ± 0.17		
	HP-βCD	2.04 ± 0.03		
RVC/SL		3.38 ± 0.05	18	28
RVC/HP-βCD		3.64 ± 0.02	22	37
RVC/HP-βCD/SL		3.14 ± 0.12	25	44
	BTB	7.70 ± 0.12		
BTB/HP-βCD		1.90 ± 0.25	98.3	72,279

RVC—Ropivacaine; BTB—Butamben; SL—Stealth liposomes; D—Diffusion Coefficients; HP-βCD—2-Hydroxypropyl-β-cyclodextrin.

**Table 3 molecules-27-04170-t003:** Data obtained in the STD experiments.

Frequency (ppm)	Area STD	Area Outside Resonance	Map STD Standardized
H2 e 6 BTB = 6.67	0.0242	0.8989	97.4%
H3 e 5 BTB = 7.66	0.0234	0.8575	97.3%
H9 BTB = 1.38	0.1550	1.0000	84.6%
H8 BTB = 1.68	0.1639	1.3054	87.5%
H14 RVC = 0.90	0.0035	1.3873	34%
H1, H2, H3 RVC = 7.14	0.0074	1.0000	100%
H15, H16 RVC = 2.14	0.0049	1.8851	35%

**Table 4 molecules-27-04170-t004:** Diameter of the vesicle liposome, polydispersity value and zeta potential.

Particles	Size (nm)	IP	PZ (mV)
SL	202 ± 1.02	0.30	−22.0 ± 0.90
BTB/SL	200 ± 0.91	0.40	−23.5 ± 0.63
BTB/HP-βCD/SL	229 ± 0.91	0.36	−25.1 ± 0.26
RVC/SL	200 ± 0.88	0.40	−22.5 ± 0.60
RVC/HP-βCD/SL	285 ± 0.91	0.40	−25.1 ± 0.26

SL—Stealth Liposome; BTB/SL—Butamben and Stealth Liposomes; BTB/SL/HP-βCD:BTB—Stealth Liposomes, 2-Hydroxypropyl-β-cyclodextrin and butamben; RVC/SL–Ropivacaine and Stealth Liposome; SL/HP-βCD/RVC Stealth liposomes, 2-Hydroxypropyl-β-cyclodextrin and ropivacaine.

## Data Availability

Not applicable.

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
