# Peer review of "Inclusion Complex between Local Anesthetic/2-hydroxypropyl-β-cyclodextrin in Stealth Liposome"

_molecules, 2022, doi:10.3390/molecules27134170_

Round 1

Reviewer 1 Report

In the present work, Souza and his/her co-workers investigated on developing a formulation for local anesthetic 85 butamben (BTB) and ropivacaine (RVC) in a binary system through its complexation in 86 2-hydroxypropyl-β-cyclodextrin (HP-βCD), followed by encapsulation in stealth lipo-87 somes. I found this paper valuable for the journal readers after some minor revisions:

1- Don't use bulk references.

2- The quality of Fig. 2 should be improved.

3- There are some grammatical mistakes. The manuscript should be revised in the term of the English language by a native person.

4- The encapsulation mechanism of drugs inside the liposome vesicle should be described in detail in the introduction section.

5- I think it's better to show the x-axis in figure 1 in the log scale.

6- The x and y font label size in figure 1 should be increased.

7- The main outcome of this research should be validated by the previous studies.

8- The Materials and Methods section should be moved before the results section.

9- results section should be renamed to "Results and discussion".

10- All the equations need references.

11- The conclusion section is presented poorly. It should be improved.

12- The following references could help you to make a good story in the introduction section for the encapsulation process:

https://doi.org/10.3390/molecules26164920

https://doi.org/10.1038/s41598-021-98222-2

doi: 10.2147/IJN.S298699

Author Response

 Thank you for your valuable suggestion.

Reviewer 2 Report

In this manuscript, the authors present very important information about the effective delivery of local anesthetics (ropivacaine and butamben) via cd-liposomal capsulated in ternary systems for prolonged anesthesia with reduced cytotoxicity. The manuscript is worth publishing in Molecules if the authors perform more experimental work. I have the following comments:

·         Authors claim sustained release of local anesthetic drugs from cyclodextrin encapsulated but however authors failed to perform drug release studies.

·         Authors have not performed any experiments to demonstrate the stability of cyclodextrin drug encapsulated liposomes. I suggest that others provide a brief description of the stability aspects of the drug encapsulated liposomes. Provide supportive references about similarly performed works.

·         Provide the slope and correlation coefficient (r2) value in Figure 1. Please also report the standard deviations of the plots.

Author Response

Thank you for your valuable suggestion.

Round 2

Reviewer 2 Report

-